# N-Terminal Acetyltransferases Are Cancer-Essential Genes Prevalently Upregulated in Tumours

**DOI:** 10.3390/cancers12092631

**Published:** 2020-09-15

**Authors:** Costas Koufaris, Antonis Kirmizis

**Affiliations:** Department of Biological Sciences, University of Cyprus, 1678 Nicosia, Cyprus

**Keywords:** pan-cancer, DepMap, CCLE, N-terminal acetylation, histones, NATs, NAA40

## Abstract

**Simple Summary:**

Cancer remains one of the leading causes of mortality globally. Ultimately, cancers are driven by the disruption of normal mechanisms that control the growth and behaviour of our cells. Improvements in our knowledge of how these normal cell control mechanisms are disrupted in cancers can potentially lead to better diagnosis and new treatments. In this study, we examined the involvement of a specific gene family, encoding protein N-terminal acetyltransferases (NATs), in various types of tumours by analysing available large-scale cancer-associated datasets. We report several novel findings relating to how NATs are disrupted in cancers highlighting specific tumours where NATs can be of biological importance and may serve as therapeutic targets.

**Abstract:**

N-terminal acetylation (Nt-Ac) is an abundant eukaryotic protein modification, deposited in humans by one of seven N-terminal acetyltransferase (NAT) complexes composed of a catalytic and potentially auxiliary subunits. The involvement of NATs in cancers is being increasingly recognised, but a systematic cross-tumour assessment is currently lacking. To address this limitation, we conducted here a multi-omic data interrogation for NATs. We found that tumour genomic alterations of NATs or of their protein substrates are generally rare events, with some tumour-specific exceptions. In contrast, altered gene expression of NATs in cancers and their association with patient survival constitute a widespread cancer phenomenon. Examination of dependency screens revealed that (i), besides NAA60 and NAA80 and the NatA paralogues NAA11 and NAA16, the other ten NAT genes were within the top 80th percentile of the most dependent genes (ii); NATs act through distinct biological processes. NAA40 (NatD) emerged as a NAT with particularly interesting cancer biology and therapeutic potential, especially in liver cancer where a novel oncogenic role was supported by its increased expression in multiple studies and its association with patient survival. In conclusion, this study generated insights and data that will be of great assistance in guiding further research into the function and therapeutic potential of NATs in cancer.

## 1. Introduction

N-terminal acetylation (Nt-Ac) is a ubiquitous eukaryotic protein modification, involving the transfer of an acetyl group from the acetyl-CoA donor to the terminal alpha-amino group of proteins [1]. This protein modification is deposited through the action of the specialised N-terminal acetyltransferase (NAT) complexes. In the archaea kingdom, there is only one NAT ortholog, which is probably the ancestor of the eukaryotic NAT genes [2]. In the eukaryotic lineage, a substantial expansion occurred, so that eight NATs are currently known, with seven of them being also present in humans [3]. The human NATs are highly evolutionary conserved and stable, being already present in the last common eukaryotic ancestor [4]. The eight NATs (NatA-NatH) are composed of at least a catalytic subunit (NAA10-NAA80) responsible for the Nt-Ac reaction. For NatA, NatB, NatC, and NatE complexes, the catalytic subunits are also associated with one or more auxiliary subunits (NAA15, NAA25, NAA35, and NAA38), which affect their binding to the ribosome or protein substrates. Human cells encode the catalytic subunits NAA10-NAA60 and NAA80, plus the associated auxiliary subunits, while NAA70 (NatG) is only found in plants [3].

Earlier studies suggested that NATs mediate Nt-Ac primarily co-translationally while bound onto ribosomes, but there is now evidence that some NATs act also post-translationally [3]. The major modifiers of the human proteome are NatA and NatB, which have been estimated to Nt-Ac 38% and 21% of all proteins respectively, while NatC, NatE, and NatF collectively act on 21% of the proteome. In contrast to this more widespread NATs, NAA40 (NatD) and NAA80 (NatH) are much more selective, with NAA40 acting on histones H2A and H4, while NAA80 acts on Actin [5,6,7]. The set of substrates targeted by each NAT is determined mainly by the protein’s initial amino acid sequence. NatA acts on proteins from which the initiator methionine has been removed through the action of methionine aminopeptidases, while NatB, NatC, NatE, and NatF act on proteins which contain the initiating methionine. NAA40 modifies histone proteins which contain the unique “SGRG” sequence, while NAA80 acts post-translationally on actin N-termini [3,8]. Mammalian cells also possess paralogues of NAA10 and NAA15, termed NAA11 and NAA16 respectively, which can also act as functional subunits of NatA [9,10].

For a long time, it was believed that Nt-Ac marks were protein modifications of limited biological relevance. However, it is now clear that, through the deposition of Nt-Ac NATs have various effects on protein function, including their folding and degradation, ability to form complexes, and their membrane and subcellular localisations [3]. Moreover, NAA40 uniquely for a NAT member can also modulate the establishment and cross-talk of histone modifications which impact chromatin structure and gene expression [11,12,13,14].

Recently, there has been an increased focus on NATs and their associated biological functions, because of emerging evidence implicating them in major human pathologies such as cancer. Different NATs have been reported to be involved in cancer e.g., NAA30 with glioblastoma [15], NAA20 in hepatocellular carcinoma [16], NAA40 in colon and lung [11,12,17] and NAA10 with various types [18]. Therefore, there already exists evidence for a role of NATs in certain human cancers, but a systematic investigation interrogating their implication across several tumour types is currently lacking.

Over the past decade, the rapid technological improvements in high-throughput molecular and other multi-omic technologies, coupled with the considerable investment into collaborative consortiums such as The Cancer Genome Atlas (TCGA) and the Cancer Cell Line Encyclopedia (CCLE) have generated a treasure trove of readily available genomic, transcriptomic, genetic dependency, and other datasets. The systematic integration of such datasets can offer important biological and medical information, for example in a recent study of 37 lysine acetyltransferases (KATs) [19]. The goal of this study was to perform a comprehensive investigation of human NATs in cancer, through a multi-omic integration of cancer datasets. Notably, the findings of this study support the widespread oncogenic potential of NATs, propose novel genomic and epigenetic carcinogenic mechanisms, and highlight new NAT-tumour connections which can be pursued further.

## 2. Results

### 2.1. Low Frequency of Genomic Alteration of NAT Genes across Human Cancers

Initially, we examined the frequency and types of genomic alterations affecting NAT genes in the TCGA pan-cancer study consisting of 32 tumour types. The Oncoprint for the NAT family (7 catalytic subunits, 5 auxiliary units, and the NAA11/NAA16 paralogues) across the TCGA pan-cancer study reveals a low genomic alteration frequency for individual NAT genes, found in 0.8–2.8% of examined tumour samples (Figure 1A). Compared to a collection of well-established cancer genes on which we performed the same analyses, genomic alterations of NATs are considerably lower (TP53—36%; PTEN—12%; KRAS—9%; MYC—9%; RB1—7%). We also compared NATs to 34 KATs, as defined in a recent report [19], which showed that the former are, on average, affected at a lower frequency (Appendix A), and several KATs were altered in 4–6% of samples (e.g., KAT6, EP300, TAF1). On average, therefore, NAT genes are characterised by a low prevalence of mutational activity across all tumour types.

Subsequently, we examined the possibility that NAT genes displayed preferential genomic alterations within specific tumour types. NAT gene DNA copy number variants (CNV) were observed at a greater than 5% frequency for only five NAT-tumour type pairs, from which only NAA40 in UCS (uterine carcinosarcoma) was above 10% (Figure 1B, left panel). Moreover, for NAA40 in UCS, NAA50 in LUSC (lung squamous carcinoma), and NAA16 in prostate adenocarcinoma (PRAD), for which a sufficient number of samples displayed gene amplifications, CNVs were associated with increased transcript levels (Figure 1B, right panel), thus supporting gene amplification being a mechanism for their transcriptional deregulation.

Next, we determined the frequency of sequence mutations within the open reading frame (ORF) of NATs in each of 32 pan-cancer tumour types. Again, we found that, in general, mutational frequency of NAT genes was rare, with only five NAT-tumour type pairs having mutational frequencies above 5%, especially involving the three auxiliary subunits NAA15, NAA25 and NAA35, and the paralogue auxiliary subunit NAA16 (Figure 1C). The increased rate of mutations for these auxiliary subunits in comparison to other NATs was evident across all tumour types (Figure 1A). To evaluate whether the increased mutation rate of these auxiliary NATs relates to their comparatively larger ORF sizes (e.g., NAA15 is 866 amino acids in length while NAA10 is 235 amino acids), the frequency of mutations was normalised to the amino acid length of each protein. Interestingly, normalisation for ORF length revealed a clear predominance of truncating—nonsense, frameshift, and splicing mutations—but not of missense mutations for the auxiliary NAT subunits, except for HYPK (Figure 1D). In terms of absolute counts, more than three-quarters of NAT truncating mutations affect just four auxiliary NATs, namely NAA15, NAA16, NAA25, and NAA35 (Figure 1E). Furthermore, it is of much interest that certain truncating NAT mutations are recurring across the TCGA pan-cancer study, especially in uterine corpus endometrial carcinoma (UCEC) (Figure 1F). Thus, truncating mutations of specific auxiliary NATs could be selectively occurring in tumours.

In conclusion, our analyses of genomic alterations of NATs in cancers found these to be relatively low frequency events, with some notable exceptions, such as NAA40 amplification in UCS, and NAA15 truncating mutations in UCEC.

### 2.2. Protein Residues Targeted by NATs Are Rarely Mutated, Apart from Histones H4 and H2A

Besides the mutation of NAT genes themselves, it is possible that mutations of residues within their protein substrates can also disrupt NAT-associated regulation in cancers. For example, a loss of Nt-Ac could potentially destabilise tumour suppressor proteins or favour more oncogenic functions. We, therefore, examined the degree to which mutations of protein residues targeted by NATs are important tumour events.

The broad-spectrum NATs act to Nt-Ac either the initial methionine (iMET) of proteins or newly exposed amino acids after the removal of iMET through the action of methionine aminopeptidases (Figure 2A). Examination of the missense mutational count for the first five amino acid residues (P1-P5) across the TCGA pan-cancer study found similar levels for each amino acid position, thus mutations of the first two amino acids that would be associated with NAT regulation are not globally enriched in cancers (Figure 2B). Since mutations of iMet are in general associated with loss-of-function [20], and this makes their effect hard to interpret, we then focused our mutational analysis only on P2. Notably, mutations of P2 in TCGA are widely but thinly distributed, with the vast majority of genes being mutated at this position in only one or two tumour patient samples (Figure 2C). Furthermore, almost a third of all identified mutations of P2 (706/2183) are likely not significant in terms of their effect on Nt-Ac, as they involve exchanges that preserve amino acid targets of NatA (e.g., serine to alanine) or NatB (e.g., aspartate to glutamate). Given that mutation data were collected from more than 9000 tumour samples, it is evident therefore that mutations of P2 are not mutational hotspots in cancer.

Unlike the broad-spectrum NATs, NAA40 is thought to specifically Nt-Ac H2A and H4 proteins which share the “SRGR” N-terminal motif (Figure 2D). Examining the presence of missense mutations within the “SRGR” motif across the TCGA pan-cancer study, we found that mutations within this motif of individual H2A/H4 genes was low. However, when the multiple gene copies of H2A and H4 are considered collectively, the former is mutated in 42 cases (close to 0.5% of all samples), and the latter in 27 cases (close to 0.3% of all samples). Moreover, there were some notable observations with respect to the affected residue and recurring mutations. First, mutations for both H2A and H4 occurred most frequently at S1 and R3, residues which cross-talk with histone N-terminal acetylation, such as S1 phosphorylation and H4R3 methylation, to impact gene expression and cell biology [11,13,14,21] (Figure 2D,E). Second, specific H2A and H4 mutations were recurring, e.g., serine 1 to cysteine and arginine 3 to glutamine for H2A (Figure 2F) (Appendix A). It is interesting to note that H2A/H4 missense mutations make up a significant proportion of all detected mutations of serine at P2 of proteins across the TCGA pan-cancer study (7%), and an even greater proportion of serine to cysteine mutations at this position (23%). These findings indicated that certain mutations of the histone motif recognised by NAA40 occur systematically and can underlie interesting cancer-associated mechanisms.

NAA80 is the second highly selective NAT, targeting the N-terminal region of the six actin isoforms [22]. Across the TCGA pan-cancer study, only three cases of missense mutations of the N-terminal amino acid of the actin isoforms were found, suggesting that this is not a significant carcinogenic mechanism.

In conclusion, mutations of protein substrates of NATs do not appear to be significant genetic alterations in cancers, with the exception of recurrent mutations of H2A/H4.

### 2.3. Gene Expression of NATs Is Widely Deregulated in Tumours

Considering the overall low genomic alteration rates of NAT genes or of their substrates in cancer, we subsequently investigated changes in gene expression as a driver of NAT cancer deregulation. For a comprehensive evaluation of NAT transcript levels in cancer, we examined RNA-Seq data from tumour and adjacent normal tissue in the TCGA pan-cancer study, setting the significance threshold at 1.5-fold-change and *p* < 0.01 (Figure 3A). To increase the reliability of the findings, analyses were performed only on data for 16 tissues, which contained at least ten normal and ten tumour samples. It is interesting to note that the upregulation of NAT genes is significantly more frequent than downregulation. The most frequently upregulated NAT gene was NAA10 (9/16 cancer types), followed by NAA20 and NAA40 (6/16 cancer types). On the opposite end of the spectrum, NAA35 and NAA11 were not significantly altered in any of the examined cancer types. For NAA11, its expression was very low or absent in the majority of samples across tumour types, due to the extensive DNA methylation of its promoter (Appendix A), consistent with previous reports [23].

To confirm our findings, we additionally examined the Oncomine database of microarray datasets, which measure transcript levels in independent samples and use a different technology than the TCGA study (Appendix A, Appendix A). This microarray database also includes tumour types for which comparisons were not available in a TCGA pan-cancer study. In agreement with the findings above, comparison of normal with tumour transcript levels revealed a general trend for increased expression of NATs in cancers. For certain NATs, it was found that multiple independent datasets corroborate the altered transcript levels observed in the TCGA pan-cancer comparison of tumour vs. normal samples, e.g., NAA10 and NAA20 in colorectal cancer and NAA40 in liver cancer. Thus, examination of the independent microarray databases confirms the notion that NATs are predominantly upregulated across multiple tumour types.

Given the finding that NATs are often deregulated at the mRNA level in cancers, we next examined the potential contribution of epigenetic deregulation to this phenotype. Altered DNA methylation was the first epigenetic dysregulation identified in cancer and is highly prevalent [24]. Therefore, we sought to find whether altered methylation of DNA could drive the altered expression of NATs between normal and tumour tissue. For this purpose, we examined the genome-wide DNA 450K array data for the TCGA pan-cancer study, focusing on the genomic region within the gene body and up to 2 kb upstream of the transcriptional start site (TSS) for each NAT gene. Our filtering criteria were set as follows: (i) DNA probes were examined in tissues where NAT transcript levels were at least 1.5 fold different between tumour and normal tissue, (ii) DNA probes with β values below 0.2 were not considered further, (iii) significant associations were deemed those with median Δβ < 0.1, correlation between transcript levels and probe methylation was <−0.2, and *p* < 0.01. Based on these criteria, altered methylations of five distinct DNA probes were anti-correlated with expression of NATs in seven tumour types (Figure 3B). Three of these probes correspond to NAA10 and one each for NAA20 and NAA40. The cg10759293 probe located within the first intron of NAA20 displayed both the most prominent decreases in its methylation (Figure 3B, right panel) and strongest anti-correlation with gene expression in two tumour types, lung squamous carcinoma (LUSC, r = −0.61) and esophageal carcinoma (ESCA, r = −0.56) (Figure 3C). Recently, it has been appreciated that methylation within the first intron has an important role in determining gene expression [25]. Of the other DNA methylation probes which display a more moderate relationship (r = −0.23 to −0.34), three (cg22971920, cg02602300, cg25758314) are located in the promoter of NAA10 and one (cg00356723) in the promoter of NAA40.

Finally, the association between NAT transcript levels and patient survival across 21 tumour types of TCGA pan-cancer were assessed, with 15 NAT-tumour associations being statistically significant. These associations involve nine different NATs and five different tumour types, including kidney cancers (KIRC and KIRP), low-grade gliomas (LGG), liver cancer (LIHC), and PAAD (Table 1). Remarkably, for 11 of these paired associations, the higher expression of NAT correlated with worst patient survival, while four correlated with improved survival. Six of the survival associations involved low grade gliomas (LGG), for which there were no normal samples in the TCGA to allow a normal-tumour comparison. However, in another published study comparing normal with glioblastoma tissues [26] (Appendix A), NAA30 and NAA80, which are associated with improved survival are downregulated, while NAA15, which is associated with worse survival, is upregulated.

Collectively, the analyses conducted here support that the gene deregulation of NATs is a common tumour event and can be correlated with disease survival. Altered DNA methylation may drive the deregulation of NATs in some tumours. Importantly, the majority of the identified NAT-cancer survival connections are novel and have not been investigated so far.

### 2.4. Cancer Cells Display Dependency on Most NAT Genes

After establishing that NATs are deregulated in various cancers, we investigated the effect of targeting these genes by examining data from cancer cell dependency screens. The cancer dependency screens are highly useful for informing pharmacological targeting strategies, as well as in providing biological insights [27,28,29]. Within the DepMap genome-wide CRISPR knockout (KO) screen consisting of 739 cell lines of diverse tissue origin [30], most NATs were essential to all or to a subset of cancer cells, with the exception of NAA60, NAA80, NAA38, NAA11 and NAA16 (Figure 4A,B). Examining the transcript levels of NATs in the cancer cell line encyclopaedia (CCLE) panel, we did not find a clear relationship between the mRNA levels and the essentiality of a given gene (Figure 4C). For example, the non-essential nature of NAA60 and NAA80 in this screen did not relate to their transcript levels, as these were similar to NAA30 and NAA40, respectively. For NAA11, its levels were close to zero in most cell lines, as observed previously across tumour samples. Consequently, there exists diversity in the dependency of cancer cells to functional NAT complexes, ranging from universally redundant to commonly essential, which does not directly correlate to the transcript levels of these genes.

In order to compare the essentiality of NAT genes in relation to other genes, we next plotted a cumulative frequency plot for all 16,363 genes examined in the DepMap study (Figure 4D). Importantly, besides the three NATs, which were not essential in this CRISPR screen, the other NATs were in the top 80th percentile of most essential genes. Top amongst them was NAA10, which was in the 99th percentile of all genes examined. Consequently, the NATs represent a gene family that is highly essential to cancer cells compared to the majority of other human genes.

For comparison purposes, we also examined the dependency of cancer cells to the 34 KATs mentioned previously. This analysis found that, compared to NATs, cancer cells can in general tolerate to a greater degree the genomic deletion of KATs compared to NATs, with a small fraction of highly essential enzymes (e.g., TAF1 and ELP3) (Appendix A).

To verify the reliability and reproducibility of the CRISPR KO data on NAT essentiality, we also examined an independent genome-wide CRISPR screen performed on 14 acute myeloid leukemia (AML) cell lines [28] (Figure 4E,F). Overall, the same patterns of dependency were observed between the two CRISPR datasets, with the exception of NAA38, which displayed a much higher gene effect in the AML dataset than in the DepMap study. This was not due to the different cancer types examined in the two datasets, since NAA38 is also redundant in AML cell lines found within the DepMap study (gene effect for AML cell lines −0.03419 ± 0.09295). The diverse results for NAA38 are therefore probably due to the technical variability between the two studies. Thus, in these two independent CRISPR KO datasets, NatA, NatB, and NatE are commonly essential to cancer cell lines, NatC and NatD are essential to some cell lines, while loss of NatF and NatH was tolerated.

By combining dependency and transcriptomic data from the CCLE project, we finally searched for evidence of functional interactions between NATs. These could be revealed in two ways (i) correlated dependencies, where the same cell lines are sensitive to genetic targeting of two NAT genes-indicating shared biological functions, and (ii) whether high levels of a NAT can reduce the effect of genetic targeting of a second NAT, indicating compensatory activity. Calculating the correlation matrix for NAT co-dependencies, we found only four significant (*p* < 0.05, Pearson’s correlation) pairwise correlations between NAT genes: NAA20-NAA25, NAA30-NAA35 and NAA30-NAA38; and NAA15-HYPK (Appendix A). Notably, these significant co-dependency profiles appear to be between subunits belonging to the same NAT complex, suggesting that the individual NAT complexes impact cancer cell viability through distinct biological processes. For the link between sensitivity to NAT targeting and transcript levels of other NATs, two significant correlations were found: High levels of NAA11 were associated with reduced sensitivity to NAA10 deletion (r = 0.74; top ranked such correlation) and of NAA16 with targeting of NAA15 (r = 0.18, ranked 16th strongest) (Appendix A). For NAA11, its ability to compensate for genetic targeting of its paralog NAA10 is consistent with previous studies [10,23], therefore validating the correctness of our analyses for identifying NAT co-dependencies and compensatory activities. For the other NAT, the examination of NAT dependency and expression indicate that the complexes act independently of each other in cancer cells.

In conclusion, the dependency data consistently show the highly essential nature of most NAT complexes in cancer cells, with the exception of NatH and NatF, which appear redundant. For NAA11 and NAA16, although cancer cells seem to tolerate their genetic targeting, they can be functionally important in their ability to compensate for their paralogue enzymes. Finally, the essential biological functions of NAT complexes appear to act through distinct biological effectors and independently of each other.

### 2.5. A Novel Oncogenic Role for NAA40 in Liver Cancer

Altogether, the above analyses have revealed several novel NAT-tumour connections that can be pursued further. As a final analysis, we focused on one of the elucidated novel NAT-tumour connections, of NAA40 with liver cancer, for three main reasons: (i) the upregulation of NAA40 in liver cancer tissue compared to normal tissue (Figure 3A), (ii) the correlation between high NAA40 transcript levels and worst survival of liver cancer patients (Table 1), and (iii) a previous report that links NAA40 to hepatic steatosis [31].

In the TCGA pan-cancer study NAA40, mRNA levels are increased 2.1-fold in tumour vs. normal tissue in liver cancers (Figure 5A, left panel). To verify the upregulation of NAA40 in normal vs. tumour liver tissue, we also examined 20 studies deposited in the GEO omnibus. In 10 out of the 20 studies, NAA40 mRNA levels were greater than 1.5-fold in this specific cancer tissue, while it was not downregulated in any of them (Figure 5A, right panel). In addition, Kaplan–Meier (KM) plots show a clear trend of worse liver cancer patient survival associated with higher NAA40 levels (Figure 5B). Hence, the upregulation NAA40 is a common event in liver cancer and is associated with worse patient survival.

Surprisingly, despite the evident upregulation of NAA40 in liver cancer, it remains the tumour tissue with the lowest NAA40 expression in the TCGA pan-cancer study (Figure 5C, left). To explain this paradox, we then examined the levels of NAA40 across the GTeX RNA-Seq data from 54 non-diseased human tissues. Likewise, with tumours, levels of NAA40 were relatively low in the normal liver tissue, being the second lowest amongst 54 human tissues, and 10-fold lower than pituitary and cerebellum (Figure 5C, right). To get a better understanding of the low hepatic expression of NAA40, we next examined how NAA40 levels are temporally altered in a developing liver obtained from embryonic C57/B6 mouse through to a young animal (Figure 5D, left) and in embryonic stem (ES) cells induced to differentiate into the hepatocyte lineage (Figure 5D, right). In both the developing liver in vivo and in the differentiating ES models, the levels of NAA40 were gradually decreasing, indicating that NAA40 is developmentally repressed in the liver lineage, but reactivated in tumour tissue.

Next, to gain insight about the biological processes with which NAA40 is associated in liver cancer, we used the SEEK tool [32] to examine genes and pathways correlating with NAA40 expression across 107 liver cancer datasets. Enrichment analysis of the 1000 genes most strongly correlating with NAA40 in liver cancer datasets identified “Cell cycle” as the top KEGG term, followed by “DNA replication” (Figure 5E). The genes within the enriched KEGG categories can be seen in Appendix A. Within the cell cycle category were included genes of the MCM complex (MCM2-7), the Anaphase promoting complex (ANAPC4, ANAPC5, ANAPC7), initiation of DNA replication (e.g., CDC45, CDC6), spindle regulation (e.g., BUB1-BUB3), and cycling proteins (e.g., CCNB1, CCNE1). These correlations suggest the involvement of NAA40 in cell cycle processes in physiological and pathological conditions. In agreement with this possibility, the NAA40 transcript levels display considerable increase in regenerating liver following 2/3 hepatectomy (Figure 5F).

Overall, this analysis points to an oncogenic role for NAA40 in liver tumorigenesis that is worth investigating in future studies.

## 3. Discussion

The increased knowledge regarding NAT biology and structure over the past decade has led to their recognition as potential molecular cancer targets [3,33]. Collectively, the multi-omic interrogation approach utilised here reaffirmed their oncogenic potential, clarified molecular drivers of their deregulation, and identified new interesting NAT-cancer connections that can be the focus of future studies.

The examination of genomic data for NATs across the TCGA pan-cancer study demonstrated their low mutational frequency, for example compared to KATs. Nevertheless, three interesting cases of genomic alterations revealed by this study were (1) NAA40 amplification in UCS, (2) NAA15 truncating mutations in Uterine CS, and (3) recurrent H2A/H4 mutations. UCS is a rare gynaecological cancer accounting for 5% of all uterine malignancies. NAA40 is within one of the 25 regions identified to be frequently amplified in UCS [34]. Why this region is amplified specifically in this tumour type and its clinical significance are not known. Regarding the recurring truncating NAA15 mutations, these have been associated previously with intellectual disability and congenital disorders [35], although not cancer. The high prevalence of truncating NAA15 mutations was somewhat surprising, given that it is also highly essential to cancer cells (Figure 4). Mutations of the S1 amino acid of histones H2A and H4 have previously been reported to be among the most frequent histone mutations [36]. Clearly, amino acid substitutions within this sequence are not random, but instead specific missense mutations are recurring, especially of S1 and R3 in both these NAA40-targeted histones. It is interesting to note that both S1C and R3C mutations are expected to impact not only NAA40 substrate recognition and Nt-Ac, but also crosstalk with other internal histone modifications (S1 phosphorylation and R3 methylation). For example, the S1C mutation would lead to a situation where NAA40 targeting is lost, but also the antagonistic phosphorylation cannot be deposited. Examining the molecular, cellular, and clinical effects of the identified NAT genomic alterations will be an interesting area for future investigations.

Unlike genomic alterations, the gene expression changes of NATs revealed through the comparison of tumour vs. normal tissues point towards their deregulation being a widespread phenomenon for several of the NAT genes (Figure 3A, Appendix A). A first observation was that the upregulation of NATs is the predominant carcinogenic effect, with few cases of downregulation. A second observation was that some NATs were deregulated in multiple tumour types (e.g., NAA10, NAA20, and NAA40), while others in few or none (e.g., NAA30, NAA60, NAA35). Consistent with the findings here, some of the NAT-tumour associations uncovered by the cancer vs. normal RNA expression comparison have been reported previously, such as NAA10 in lung, colorectal, liver, and breast cancer [18,19], NAA20 in liver [16]. We have also shown potential mechanistic links between altered DNA methylation and the deregulation of NATs in some tumours. Importantly, most of the reported NAT-tumour associations have been unveiled in this study for the first time. It should also be noted that, in this study, we have focused on bulk normal vs. tumour comparisons; consequently, we may have missed important associations between specific molecular or histological tumour subtypes and NATs.

The widespread deregulation of NATs in cancer indicates that this class of enzymes could be considered as potential therapeutic targets. Indeed, previous studies have identified inhibitors/binders of NATs [37,38]. Examination of dependency screens showed that cancer cells are dependent on most NATs, and thus could be vulnerable to their targeted inhibition (Figure 4). The greatest dependency was observed for the NatA, NatB, and NatE complexes that are also known to Nt-Ac the majority of the human proteome [3]. However, it is possible that these NAT complexes are also essential to non-cancer cells as well, given their broad substrate specificity, hence they could have a narrow therapeutic window. The exception to the essentiality of NATs to cancer cells were NatF (NAA60) and NatH (NAA80), the targeting of which was tolerated by cancer cells. These two NATs perform what are currently considered to be specialised biological functions. NAA60 is mainly localized to the cytosolic side of the Golgi apparatus membranes where it acetylates transmembrane proteins, both Golgi-associated proteins and proteins of other parts of the secretory pathway [39]. NAA80 specifically mediates Nt-Ac of Actin proteins. Nevertheless, in two independent CRISPR dependency datasets, the targeting of these NATs did not affect the viability or growth of cancer cells. The non-essential nature of NatF and NatH could be due to a so far unrecognised redundancy in their functions with other NATs. Alternatively, the cell phenotypes associated with loss of these NATs could be more subtle, and not observable in the tested cell culture environments. Finally, NAA30 and NAA40 were essential to some, but not all, cancer cells, indicating that their targeting could be particularly effective in specific genetic or transcriptomic contexts.

Using the DepMap data, we also examined co-dependencies and compensatory relations between NATs. However, we did not find any evidence for such functional relationships between NAT complexes, suggesting that they are impinging upon distinct cellular processes. These observation is consistent with NAT complexes and subunits possessing distinct co-translational, post-translational, and non-canonical functions [3,40]. Notably though, we did find supporting evidence for the capacity of NAA11 to compensate for its paralog NAA10. The examination of CCLE and TCGA pan-cancer data confirmed previous reports that NAA11 is epigenetically repressed in most tissues [23]. However, in the minority of cell and tissue samples where NAA11 escapes the epigenetic silencing to be sufficiently expressed, it can partially compensate for NAA10. The sensitivity of cancer cells to pharmacological targeting of NAA10 could therefore be lower in tumours, such as testicular germ cell carcinoma, where NAA11 is consistently present [23]. Moreover, the adaptive upregulation of NAA11 could be a mechanism for cancer cells to reduce the toxicity of NAA10 inhibition.

NAA40 has recently been reported to have oncogenic functions in lung and colorectal cancers [11,12,17]. Our multi-omic analyses conducted here revealed a complex and interesting biology for NAA40, involving transcriptional deregulation, copy number amplification, and mutations of its targeted histone proteins (Figure 1, Figure 2 and Figure 3). Moreover, NAA40 was shown to be deregulated and potentially be involved in a greater diversity of human cancers than recognised so far. The observation that NAA40 is upregulated in several but not all tumour types, and that it is essential to a subset of all cancer cells, indicate that its oncogenic functions are complex, and vary according to the genetic and/or epigenetic background. In addition, we chose to focus on liver cancer as a particularly interesting case where NAA40 could be of biological and clinical significance. Given the low levels of expression of NAA40 in normal liver, the potentially druggable nature of NAA40, the increasing incidence of liver cancers in the developed world, and the lack of efficacious cures for liver cancer, further investigations into the cancer role of this NAT in liver cancer are warranted.

## 4. Materials and Methods

### 4.1. Genomic Alterations

Cancer genomic alteration data for TCGA pan-cancer study were obtained from cBiolportal [41]. Mutational data for the TCGA pan-cancer panel were obtained from mc3 V02.8 public version through Xena UCSC (https://xena.ucsc.edu/).

### 4.2. Transcriptomic Analyses

RNA-Seq and DNA methylation data from tumour and adjacent normal tissues were obtained using the UCSC Xena tool [42]. Further transcriptomic data from array hybridisation studies were analysed using the Oncomine portal [43]. Information on the microarray studies deposited in Oncomine with significant alterations in NATs can be seen in Appendix A. Transcript comparisons were performed only for studies with at least 10 normal and 10 tumour samples. The association of NAT transcripts with cancer patient survival in TCGA pan-cancer tool was examined in Oncolnc database [44]. Significant associations were considered those with FDR corrected *p* values < 0.05. The expression of the NAT family members in normal human adult tissues was v8-2017 of the GTEx consortium [45].

### 4.3. Dependency Data

CRISPR (Avana Public 20Q1) and transcriptomic (Expression Public 20Q1) data for cancer cell lines were obtained from DepMap portal (https://depmap.org/portal/). For Wang et al., dependency data were obtained as Appendix A from their manuscript.

### 4.4. DNA Methylation Analyses

DNA methylation data used were β-values for CpG probes (DNA methylation 450 K) contained within NAT gene bodies, or within 2 Kb region upstream of the gene. Only probes with mean β-values > 0.2 in either normal or tumour tissue were considered for further analysis. The criteria for discovering DNA methylation regions associated with altered transcript levels were the following: (1) mean difference in the mean β-values for the probe between normal and tumour samples of at least 0.1 and *p* < 0.05, (2) Pearson’s correlation between methylation score and transcript levels of at least r = −0.2, and (3) transcript levels had to differ at least 1.5-fold between tumour and normal tissue.

### 4.5. Gene Expression Studies

Gene expression data were obtained from GEO omnibus using the GEO2R tool (https://www.ncbi.nlm.nih.gov/geo/geo2r/). When multiple probes for NAA40 were present in a dataset, the probe with the highest value was chosen. For liver analyses, the following studies were examined: Liver regeneration data after hepatectomy (GSE63742); developing mouse liver (GSE13149); differentiating ES (GSE19044). No further normalisation of expression data was performed.

## 5. Conclusions

As this study demonstrates, investigations into the cancer biology of NATs have so far only scratched the surface. There is therefore considerable novel knowledge to be extracted by future research into the cancer biology of this gene family. Using a multi-omics approach, we have uncovered, in this study, several novel interesting NAT-tumour connections, such as: tumour specific amplifications of NAA40 and NAA15 truncating mutations; recurring mutations of H2A/H4 NAA40 motif; strong anti-correlations between NAA20 first intron methylation and its expression in LUSC and ESCA tumours; ability of NAA11 to compensate for targeting of its homolog NAA10; and an oncogenic role for NAA40 in liver cancer. These newly unveiled leads can be further exploited to delineate the contribution of NATs in cancer and assess their therapeutic potential.

## Figures and Tables

**Figure 1 cancers-12-02631-f001:**
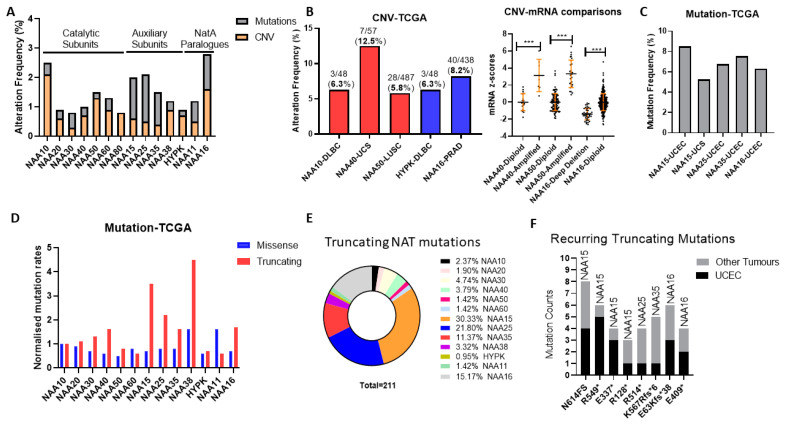
Genomic alteration of N-terminal acetyltransferase (NAT) genes across The Cancer Genome Atlas (TCGA) pan-cancer study of 32 cancer types. (**A**) Oncoprint displaying average frequency of copy number variants (CNV) and mutations for each NATs across cancer types; (**B**) Tumour types with greater than 5% frequency of NAT amplifications or deletions (left panel) and association between amplification and mRNA levels in tumours (right panel). DLBC—Lymphoid Neoplasm Diffuse Large B-cell Lymphoma, UCS—Uterine Carcinosarcoma, LUSC—Lung squamous cell carcinoma; (**C**) Tumour types with greater than 5% frequency of NAT mutations. UCEC—Uterine Corpus Endometrial Carcinoma; (**D**) counts of missense and truncating mutations for NAT genes across TCGA Pan-Cancer study normalised to protein length. Mutation counts for each NAT gene were normalised first to the length of the associated ORF and then to the corresponding value of NAA10. For NAA80, mutation data were not available; (**E**) Pie-chart depicting the frequency of truncating mutations for every NAT gene across the TCGA pan-cancer study; (**F**) Mutational count of recurring (at least three occurrences) truncating mutations of NAT genes in UCEC and other tumour types. *** *p* < 0.001 Student’s t-test.

**Figure 2 cancers-12-02631-f002:**
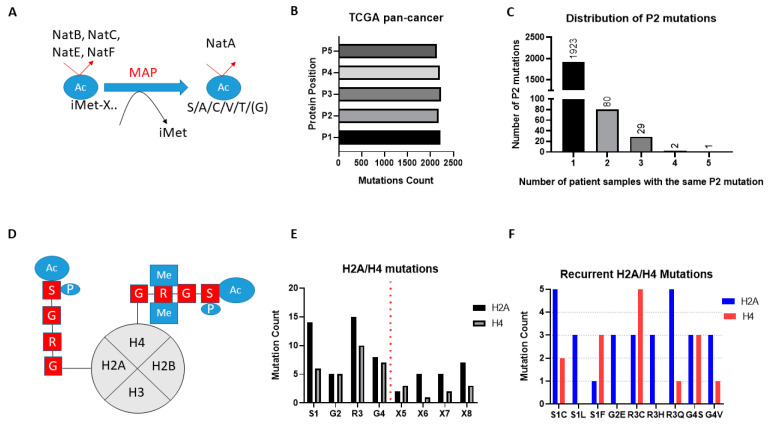
Mutations affecting NAT protein substrates across 32 cancer types. (**A**) Schematic of Nt-Ac of broad-spectrum NATs. NatB, NatC, NatE, NatF directly methylate the iMET of substrate proteins. The substrate specificity of these NATs is determined by the amino acid adjacent to the methionine. After removal of iMET proteins through the action of MAP, N-termini with the first amino acid being S/A/C/T/V/G can be Nt-Ac through the action of NatA; (**B**) Missense mutational count for first five amino acids (P1–P5) across TCGA pan-cancer; (**C**) Distribution of missense mutations of P2 of proteins across TCGA pan-cancer; (**D**) Schematic of NAA40 recognition and Nt-Ac of histone H2A and H4; (**E**) Total missense mutational count of the first four amino acids (SRGR) across TCGA pan-cancer study; (**F)** Counts of recurring missense mutations (at least two occurrences) affecting the SRGR sequence in H2A or H4.

**Figure 3 cancers-12-02631-f003:**
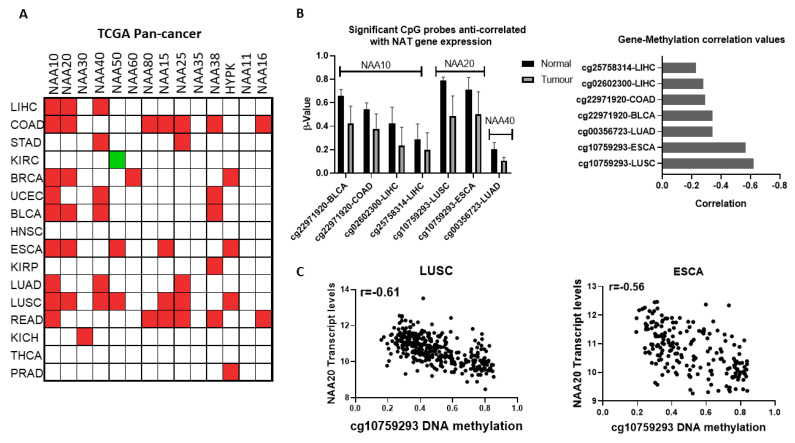
Transcriptional and epigenetic deregulation of NAT genes between normal and tumour tissues; (**A**) Comparison of NAT transcript levels in tumour and adjacent normal tissue from TCGA pan-cancer study. Significantly (*p* < 0.05) upregulated transcripts are depicted as red boxes, downregulated transcripts are depicted as green colour boxes; (**B**) DNA methylation probes displaying significant alterations in their methylation levels and anticorrelation with NAT transcript levels in Bladder Urothelial Carcinoma (BLCA), Colon adenocarcinoma (COAD), Liver hepatocellular carcinoma (LIHC), Lung squamous cell carcinoma (LUAD), and Oesophageal carcinoma (ESCA) tumours; (**C**) Plots of NAA20 transcript levels (*Y*-axis) vs. DNA methylation levels for probe cg10759293 (*X*-axis) in LUSC and ESCA tumours.

**Figure 4 cancers-12-02631-f004:**
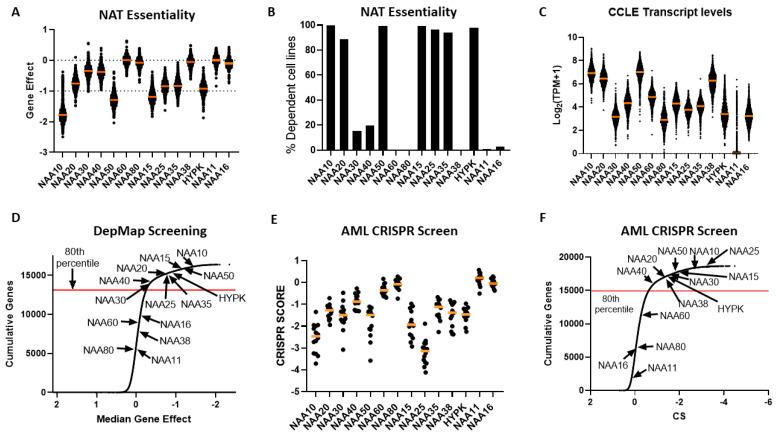
Dependency of cancer cells on functional NAT complexes. (**A**) Gene effect (CEREC) of CRISPR-CAS KO of human NAT catalytic and auxiliary genes across 739 cell lines. More negative values indicate greater sensitivity to KO. Each cell line is represented by a circle symbol, and the median gene effect across all cell lines is depicted by an orange line; (**B**) Percentage of cell lines with CRISPR-CAS KO data which were deemed as dependent on NAT gene, as calculated in DepMap database; (**C**) Transcript levels for NAT genes across CCLE study. The orange line represents the median of transcript level across the CCLE panel; (**D**) Cumulative frequency for median gene effect for all examined genes in DepMap CRISPR-CAS KO dataset. Arrows indicate the position of NAT catalytic subunits on this curve; (**E**) Scatterplot of CRISPR scores for NAT catalytic genes in study of 14 Acute Myeloid Leukaemia (AML) cell lines. Each circle symbolises a cell line and orange line depicts median CRISPR score. Lower CRISPR scores indicate greater sensitivity to KO; (**F**) Cumulative frequency for median gene effect for all examined genes in AML CRISPR KO dataset. Arrows indicate the position of NAT catalytic subunits on this curve.

**Figure 5 cancers-12-02631-f005:**
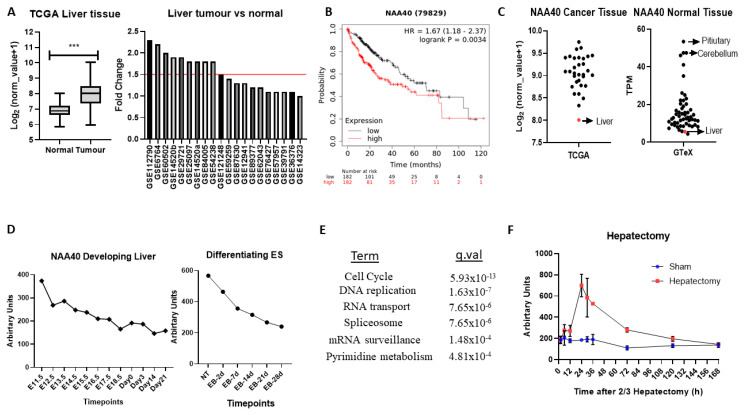
NAA40 is associated with liver proliferation and cancer. (**A**) NAA40 transcript comparison between normal and tumour liver tissue in TCGA (left panel) and 20 studies from GEO omnibus (right panel); (**B**) KM plot for liver cancer, with samples divided according to median NAA40 expression; (**C**) Transcript levels of NAA40 in TCGA pan-cancer (left panel) and GTeX study (right panel). Each circle represents mean transcript value of one tissue type, with arrow indicating relative levels of liver; (**D**) Timecourse of NAA40 transcript levels in developing liver (left panel) and in Embryonic Stem (ES) cells differentiating to hepatocytes (right panel); (**E**) KEGG enriched terms among 1000 top co-expressed genes in 107 liver cancer datasets (*q* < 0.001); (**F**) Time course of NAA40 transcript levels in regenerating liver following 2/3 hepatectomy. *** *p* <0.001 Student’s t-test.

**Table 1 cancers-12-02631-t001:** NAT transcripts significantly associated with patient survival.

Gene	Tumour	Co × Coefficient	FDR Corrected *p* Value
NAA10	KIRC	0.354	1.63 × 10^−4^
NAA20	LGG	0.235	2.06 × 10^−2^
NAA30	LGG	−0.255	1.78 × 10^−2^
NAA30	KIRC	−0.24	1.38 × 10^−2^
NAA40	KIRC	0.371	6.67 × 10^−5^
NAA40	LIHC	0.318	3.54 × 10^−2^
NAA50	LGG	0.33	1.76 × 10^−3^
NAA50	KIRP	0.486	2.75 × 10^−2^
NAA50	PAAD	0.514	4.20 × 10^−3^
NAA80	LGG	−0.264	7.56 × 10^−3^
NAA80	KIRC	0.208	2.66 × 10^−2^
NAA15	LGG	0.282	1.26 × 10^−2^
NAA25	KIRC	0.368	2.57 × 10^−5^
NAA25	KIRP	0.534	1.27 × 10^−2^
HYPK	LGG	−0.269	1.71 × 10^−2^

Corrected *p* value < 0.05 was used as threshold for significance.

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
