# Peer review of "N-Terminal Acetyltransferases Are Cancer-Essential Genes Prevalently Upregulated in Tumours"

_cancers, 2020, doi:10.3390/cancers12092631_

Round 1

Reviewer 1 Report

The manuscript ‘N-Terminal Acetyltransferases are cancer-essential genes prevalently upregulated in tumours’ by Costas Koufaris and Antonis Kirmizis is an omics investigation of databases on the links between NAT genes and cancers. There is a particular focus on NAA40 which is the key interest of this research group.

The study presents novel data of interest to the cancer community and the communities working on protein modifications, in particular N-terminal acetylation.

Overall, the data are presented properly and the manuscript is well written. However, some points should be addressed.

Suggestions for improvements:

  1. Since the analysis is meant to include all NAT subunits in human cancers, the authors should also include NAA11 and NAA16 in their general analysis (and represented in several figures and tables) since these are functional subunits in human cancers that may carry out NatA type N-terminal acetylation along with their paralogues NAA10 and NAA15.

  1. A recent publication presented a metagenomic analysis of 37 different acetyltransferases, including NAT subunits NAA10, NAA50 and NAA60 (PMID: 32162442). Please discuss current findings in relation to the data in this study.

  1. Line 31: eight NAT complexes -> eight NATs

  1. Line 34-35: ‘The eight NAT complexes (NatA-NatF) are composed of at least a catalytic subunit (NAA10-NAA80) responsible for the Nt-Ac reaction.’

-> Please change this sentence for correctness. Eight NATs = NatA-NatH, not NatA-NatF. Also, there are not 8 NAT complexes, but 8 NATs in eukaryotes (as far as we know NAA40, NAA60 and NAA70 act alone, not in complexes).  

  1. Line 43: ‘while NatC, NatE, and NatH collectively act on 21% of the proteome’

-> Please change NatH to NatF.

  1. Line 45: ‘NAA40 acting on histones H2A and H4, while NAA80 acts on Actin’

-> For such specific statements, reference to original literature is appropriate, in this case PMID: 12915400 for NAA40 and PMID: 29581253 & PMID: 30028079 for NAA80.

  1. Line 45-46: ‘The set of proteins targeted by each NAT is determined mainly by the protein’s initial amino acid sequence.’

-> Replace ‘protein’ with ‘substrate’ to avoid confusion between substrates and NATs.

  1. Figure 1B legend and elsewhere, please include abbreviation explanations, for instance DLBC, LUSC, UCS.

  1. Figure 1A and elsewhere: it would be useful to provide data for a couple of well known oncogenes to allow for comparison.

  1. Line 109-111: ‘In terms of absolute counts, more than three-quarters of NAT truncating mutations are affecting just three auxiliary NATs, namely NAA15, NAA25, and NAA38 (Fig.1E).’

-> Typo: NAA38 should be NAA35.

  1. Line 391-393: ‘These two NATs perform what are currently considered to be specialised biological functions, Nt-Ac of Golgi-associated proteins and of actin isoforms respectively.’

-> NAA60 is mainly localized to the cytosolic side of the Golgi apparatus membranes where it acetylates transmembrane proteins, both Golgi proteins and proteins of other parts of the secretory pathway (PMID: 25732826). Please rephrase accordingly.

Author Response

The manuscript ‘N-Terminal Acetyltransferases are cancer-essential genes prevalently upregulated in tumours’ by Costas Koufaris and Antonis Kirmizis is an omics investigation of databases on the links between NAT genes and cancers. There is a particular focus on NAA40 which is the key interest of this research group.

The study presents novel data of interest to the cancer community and the communities working on protein modifications, in particular N-terminal acetylation.

Overall, the data are presented properly and the manuscript is well written.

We thank the reviewer for the recognition of the value and aims of this work.

However, some points should be addressed.

Suggestions for improvements:

  1. Since the analysis is meant to include all NAT subunits in human cancers, the authors should also include NAA11 and NAA16 in their general analysis (and represented in several figures and tables) since these are functional subunits in human cancers that may carry out NatA type N-terminal acetylation along with their paralogues NAA10 and NAA15.

We have now included data on NAA11 and NAA16 in the relevant sections (Fig.1, Fig.3, Fig.4, Fig.S3).

  1. A recent publication presented a metagenomic analysis of 37 different acetyltransferases, including NAT subunits NAA10, NAA50 and NAA60 (PMID: 32162442). Please discuss current findings in relation to the data in this study.

Thank you for pointing this very recent integrated study of lysine acetyltransferases.  These study was useful for us because (a) it is another example of a study using similar approaches to ours to examine KATs (b) allows us to compare NAT mutation data to those of KATs (c) also this recent study includes analysis of some NATs, as suggested by the reviewer, allowing comparison with our analysis. We have included such comparisons to enrich our manuscript (Figs S1, S3).

  1. Line 31: eight NAT complexes -> eight NATs

Manuscript text has been changed 

  1. Line 34-35: ‘The eight NAT complexes (NatA-NatF) are composed of at least a catalytic subunit (NAA10-NAA80) responsible for the Nt-Ac reaction.’

-> Please change this sentence for correctness. Eight NATs = NatA-NatH, not NatA-NatF. Also, there are not 8 NAT complexes, but 8 NATs in eukaryotes (as far as we know NAA40, NAA60 and NAA70 act alone, not in complexes).  

 We agree with the Reviewer’s opinion. The text has been changed accordingly.

  1. Line 43: ‘while NatC, NatE, and NatH collectively act on 21% of the proteome’

-> Please change NatH to NatF.

 Thank you for pointing out this typo which has been corrected.

  1. Line 45: ‘NAA40 acting on histones H2A and H4, while NAA80 acts on Actin’

-> For such specific statements, reference to original literature is appropriate, in this case PMID: 12915400 for NAA40 and PMID: 29581253 & PMID: 30028079 for NAA80.

References have been added. 

  1. Line 45-46: ‘The set of proteins targeted by each NAT is determined mainly by the protein’s initial amino acid sequence.’

-> Replace ‘protein’ with ‘substrate’ to avoid confusion between substrates and NATs.

Change has been made

  1. Figure 1B legend and elsewhere, please include abbreviation explanations, for instance DLBC, LUSC, UCS.

This information has been now added.

  1. Figure 1A and elsewhere: it would be useful to provide data for a couple of well known oncogenes to allow for comparison.

We have added to our manuscript comparisons with known cancer genes (MYC, TP53, PTEN, RB1, KRAS)

  1. Line 109-111: ‘In terms of absolute counts, more than three-quarters of NAT truncating mutations are affecting just three auxiliary NATs, namely NAA15, NAA25, and NAA38 (Fig.1E).’

-> Typo: NAA38 should be NAA35.

Thank you for pointing out this typo.

  1. Line 391-393: ‘These two NATs perform what are currently considered to be specialised biological functions, Nt-Ac of Golgi-associated proteins and of actin isoforms respectively.’

-> NAA60 is mainly localized to the cytosolic side of the Golgi apparatus membranes where it acetylates transmembrane proteins, both Golgi proteins and proteins of other parts of the secretory pathway (PMID: 25732826). Please rephrase accordingly.

This has been rephrased accordingly in the manuscript.

Reviewer 2 Report

The authors have investigated whether the expression of N-terminal acetyltransferase (NAT) complexes are correlated with tumors or their progression. This manuscript shows that NAA40 mutation was recruited in histone H2A and H4 with NAA15 in tumors, and NAA40 involved transcriptional degradation in liver. It was impressive that the authors investigated various type of tumors whether NATs affect mutational activity, and show specific methylation using a multi-omic integration datasets.

Comment:

  • Since the authors used various tumor samples, reviewer ask to consider the correlation of NAA transcriptional changes on each tumor or their histological types.
  • In Figure 5E, NAA40 expression in liver cancer detasets correlated with cell cycle and DNA replication. The authors described just only the word, for example “cell cycle”. The detailed explanation or discussion about pathways, mutation, or strongly related genes are needed.
  • It might put the cart before the horse between Figure 4 and Figure 5. The description of Figure 4 was presented in “Discussion” for the first time.

Author Response

The authors have investigated whether the expression of N-terminal acetyltransferase (NAT) complexes are correlated with tumors or their progression. This manuscript shows that NAA40 mutation was recruited in histone H2A and H4 with NAA15 in tumors, and NAA40 involved transcriptional degradation in liver. It was impressive that the authors investigated various type of tumors whether NATs affect mutational activity, and show specific methylation using a multi-omic integration datasets.

Comment:

  • Since the authors used various tumor samples, reviewer ask to consider the correlation of NAA transcriptional changes on each tumor or their histological types.

If we understand correctly here the reviewer is asking us to consider the correlations of NATs with molecular or histological subtypes of tumours. Although indeed this kind of tumour-specific analyses can be very important, it is also considerable work that falls beyond the scope of this study. We have added a sentence in Discussion relating to this.

  • In Figure 5E, NAA40 expression in liver cancer detasets correlated with cell cycle and DNA replication. The authors described just only the word, for example “cell cycle”. The detailed explanation or discussion about pathways, mutation, or strongly related genes are needed.

In accordance with the Reviewer’s suggestion we have now added to our manuscript Supplementary Table 5 which includes the names of the genes found within the enriched KEGG categories and mentioned in our manuscript some examples of the significantly correlated genes.

  • It might put the cart before the horse between Figure 4 and Figure 5. The description of Figure 4 was presented in “Discussion” for the first time.

The reviewer is correct in this matter, which related to PDF preparation from the uploaded manuscript. These will be corrected in the final published version.

Reviewer 3 Report

Thank you for the opportunity to review the manuscript "N-terminal Acetyltransferases are cancer-essential genes prevalently upregulated in tumours" by Koufaris and Kirmizis.

In this manuscript the authors show, using a variety of publicly available cancer datasets that N-terminal acetyltransferases (NAT), while having a relatively low frequency of mutations across multiple cancer types, have higher mutation rates in specific cancers including uterine corpus endometrial cancer and  uterine carcinosarcoma. They go on to show using genome-wide CRISPR knockout database information that NAT complexes except NatH and NatF are required for cancer cell growth in a wide variety of cancer cell lines. Based on these cell line results the authors go on to show a role for NAA40 in liver cancer based on overexpression in tumour vs normal liver tissue and worse survival of patients with high NAA40 expression.

While the authors do not propose or show a mechanism for the effects, they clearly show concordant results from independent sources sufficient to justify further investigations into the role of NAT in cancer, particularly liver cancer. Future prospective studies to test mechanistic hypotheses are required to develop the ideas presented in the current manuscript. However, if feasible these would increase the impact of the current manuscript.

The only changes required are a number of small formatting issues potentially related to PDF conversion of the source documents during manuscript submission - Figures 5 precedes Figure 4 based on the current page layout and Figure S2D and S2E did not seem to be included in the supplemental data file.

Finally, The figures referenced on lines 243, 244, 250 and 257 (all referenced panels of Figure 3) seem to relate to data shown in equivalent panels of  Figure 4.

Author Response

Thank you for the opportunity to review the manuscript "N-terminal Acetyltransferases are cancer-essential genes prevalently upregulated in tumours" by Koufaris and Kirmizis.

In this manuscript the authors show, using a variety of publicly available cancer datasets that N-terminal acetyltransferases (NAT), while having a relatively low frequency of mutations across multiple cancer types, have higher mutation rates in specific cancers including uterine corpus endometrial cancer and  uterine carcinosarcoma. They go on to show using genome-wide CRISPR knockout database information that NAT complexes except NatH and NatF are required for cancer cell growth in a wide variety of cancer cell lines. Based on these cell line results the authors go on to show a role for NAA40 in liver cancer based on overexpression in tumour vs normal liver tissue and worse survival of patients with high NAA40 expression.

While the authors do not propose or show a mechanism for the effects, they clearly show concordant results from independent sources sufficient to justify further investigations into the role of NAT in cancer, particularly liver cancer. Future prospective studies to test mechanistic hypotheses are required to develop the ideas presented in the current manuscript. However, if feasible these would increase the impact of the current manuscript.

We thank the reviewer for recognising the contribution of this work and its future perspectives.

The only changes required are a number of small formatting issues potentially related to PDF conversion of the source documents during manuscript submission - Figures 5 precedes Figure 4 based on the current page layout and Figure S2D and S2E did not seem to be included in the supplemental data file.

Thank you for pointing this out. The placement of Figures 4 and 5 will be corrected in the final manuscript.

Regarding the mention to Fig. S2D and S2E, these were mistakenly included during manuscript preparation. The reference to them in Figure legend have been removed accordingly.

Finally, the figures referenced on lines 243, 244, 250 and 257 (all referenced panels of Figure 3) seem to relate to data shown in equivalent panels of  Figure 4.

Thanks for pointing out this error which has now been corrected.

Round 2

Reviewer 1 Report

All comments have been appropriately addressed. 

Reviewer 2 Report

The authors have investigated that he expression of N-terminal acetyltransferases (NATs) are correlated with tumors or their progression and they show an important role of NAA40 expression in liver cancer. The authors answered to all comments and added the accurate information, referring the recent reports, explanation of results in detail, and discussion. This manuscript is become better understand. The reviewer hope that the authors would report the therapeutic potentials of NAT in various cancers and correlation with overall survivals in the future.

Reviewer 3 Report

Thank you for the opportunity to review the updated version of the manuscript manuscript "N-terminal Acetyltransferases are cancer-essential genes prevalently upregulated in tumours" by Koufaris and Kirmizis.

The authors have clearly addressed my specific minor formatting and typographical issues. The additional analysis including comparison with KAT data suggested by the other reviewer also strengthens the manuscript.